# C-GATS: Conditional Generation of Anomalous Time Series

**Vikramank Singh**
AWS AI Labs
vkramas@amazon.com

**Abishek Sankararaman**
AWS AI Labs
abisanka@amazon.com

**Balakrishnan (Murali) Narayanaswamy**
AWS AI Labs
muralibn@amazon.com

**Zhao Song**
AWS AI Labs
songzhao@amazon.com

## Abstract

Sparsity of the data needed to learn about anomalies is often a key challenge faced when training deep supervised models for the task of Anomaly Detection (AD). Generating synthetic data by applying pre-determined transformations that conform to a set of known invariances has shown to improve performance of such deep models. In this work we present C-GATS to show that one can *learn* a much larger invariance space using the available sparse data by training a conditional generative model to do Data Augmentation (DA) for anomalous Time Series (TS) in a model-agnostic way. Particularly, we factorize an anomalous TS sequence into 3 attributes— *normal sub-sequence*, *anomalous sub-sequence*, and *position of the anomaly* and model each of them separately. This factorization helps exploit samples from the *dominant* class i.e normal TS to train a generative model for the *sparse* class i.e anomalous TS. We provide an exhaustive study to showcase that C-GATS not only learns to generate different types of anomalies (eg: point anomalies and level-shift) but those generated anomalies improve performance of multiple SOTA TS AD models on a set of popular public TS AD benchmark datasets.

## 1 Introduction

Deep models have been successful at achieving exceptional performance on a wide variety of machine learning tasks, ranging from vision and natural languages understanding to TS forecasting and anomaly detection [24]. The models have millions or even billions of parameters, allowing them to learn and represent complex functions, features and transformation automatically from data. However, learning the values of these parameters requires a large data-set. Gathering real world labelled training data is usually non-trivial and a costly task. As a result, the performance of deep algorithms is limited in data sparse tasks such as segmentation or AD [37].

A well studied way to handle this problem is by generating synthetic data that belongs to the original data distribution [28, 9, 10, 18, 34, 15]. We categorize these methods into two broad categories-(i) when invariances are known *apriori*; (ii) when invariances are *learned* from the original data distribution. The traditional Data Augmentation (DA) techniques like scaling, translations, rotations and their variants fall in the former category [13, 43, 40, 19] while the generative approaches like Generative Adversarial Networks (GANs), Variational Auto Encoders (VAEs) and their variants fall in the latter category. Although considered best practices for a long time [30], the former class of methods are shown to have limitations such as incapability of exploring a large invariance space [1]

NeurIPS 2022 Workshop on Synthetic Data for Empowering ML Research.

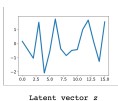

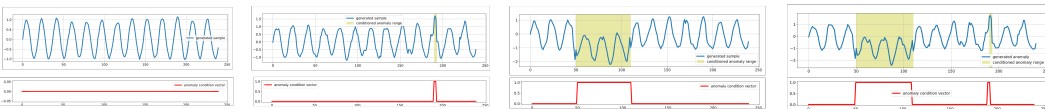

Figure 1: For a fixed latent $z$ sampled from a standard gaussian, we vary the conditional vector $Y^A \in \mathbb{R}^T$ (in red) and generate the anomalous TS, $X^A \in \mathbb{R}^T$. Without any additional information about the type of anomaly, `C-GATS` uses the conditional vector to infer the type of anomaly from the training data. For example, lengthier anomalies are level-shift and the shorter ones are point anomalies. `C-GATS` is able to learn useful invariances hidden in the data.

and requiring domain knowledge to design augmentation schemes [18]. Hence, in this paper we build upon the latter class of methods.

Some recent works [14, 39, 8] have shown the value of using generative models for TS data but only on TS classification task. No work has been done to explore the same for more complicated task of TS AD, the only exception being [6, 7] where the problem of AD was again framed as a binary classification task. We consider the task of univariate TS AD in its original form [37] where given a TS sample $X_i \in \mathbb{R}^T$, goal is to output a corresponding label $Y_i \in \{0,1\}^T$ where 0 denotes normal value and 1 denotes anomaly at each timestamp. In this task a well-known problem is class imbalance between the TS samples that are normal, i.e without any anomalies v/s the ones that contain anomalies, which prevents us from training deep AD models in a supervised manner [5, 17].

We argue that generating synthetic anomalous TS samples for AD is non-trivial as compared to other tasks like classification due to 3 key reasons– **(a)** *temporal labels*: the labels are no longer scalar and vary temporally for a given TS sample; **(b)** *data imbalance*: the number of TS samples in the dataset that contain anomalies are sparse; **(c)** *data scarcity*: for a given TS sample the number of data points that are anomalous are sparse. We show an example of (a) in supp figure 5 where either the DA method itself introduces new anomalies in the data or it corrupts the anomalous samples leading to contradicting labels. On the other hand, off-the-shelf deep generative models like vanilla GANs and VAEs fail to learn a good distribution of anomalous TS due to reasons (b) and (c). We show this in our experiments (Section 4). Conditional variants of these generative models [32] show some improvement by learning class dependent latent distributions, however when applied directly in our setting, face the *curse of dimensionality* [12] as the number of classes in TS AD task grow exponentially with the length of TS samples. A thorough survey of related work is provided in supp section 6.1.

To address these issues, we propose `C-GATS`, a conditional generative model trained in a semi-supervised manner to produce synthetic TS that contain anomalies. The main contribution of this paper is that these synthetic anomalies are—

- *effective*: they improve performance of several SOTA TS AD models on downstream task of AD.
- *diverse*: `C-GATS` learns to generates different types of anomalies present in the dataset (eg: point, level-shift).
- *label-preserving*: they conform to the label used to condition `C-GATS` during generation.
- *model agnostic*: they are independent of the downstream AD model.
- *domain agnostic*: we show `C-GATS` can be trained on TS dataset from multiple domains.

## 2   Problem Definition

Consider a setting where $X^A \in \mathbb{R}^T$ is a univariate TS of length $T$ with anomaly. We represent $X^N \in \mathbb{R}^T$ as a univariate TS of length $T$ without any anomaly. $Y^A \in \{0,1\}^T$ is the corresponding binary label vector for $X^A$ and $Y^N \in \{0\}^T$ is the label vector for $X^N$. We thus have a dataset $\mathcal{D}$ which comprises of $m$ number of anomalous pairs $(X^A, Y^A)$ and $n$ number of normal pairs

$(X^N, Y^N)$ such that $m << n$. Our goal is to learn a model that can produce anomalous TS with labels, i.e, learn a density $\hat{p}(X^A, Y^A)$ that best approximates $p(X^A, Y^A)$ .

## 3 Proposed Algorithm

A standard way of learning the joint distribution $(X^A, Y^A)$ is by trying to learn the conditional $p(X^A|Y^A)$ typically modeled using a CVAE [32]. This works well when the number of unique classes are relatively fewer, eg: MNIST [25] but doesn't scale as number of classes increase, i.e the condition vector becomes high dimensional [12]. In our setting, there are $2^T$ unique classes as $Y^A \in \mathbb{R}^T$ and a vanilla CVAE fails to learn a good distribution w.r.t each class due to the well-known problem in generative models called mode-collapse [33]. An example of this is seen in Figure 6.a where a vanilla CVAE fails to reconstruct an anomalous TS.

Note that in multi-class generative problems with class imbalance in vision domain, where for eg. the dominant class is *'dog'* and sparse class is *'ship'*— the two classes do not share underlying structural similarity. Hence, trying to generate samples of *'ship'* by exploiting samples from *'dog'* is possible, [4] but limited. However, in our setting the normal and anomalous TS share significant structural similarity that can be exploited. More formally, we study anomalous TS $X^A$ where we can factorize them as shown in Eq. 1 where $x^A$ is a sub-sequence of $X^A$ that contains the anomalous data points, $x^N$ is the normal sub-sequence such that $x^A \cup x^N \equiv X^A$, and $Y^A$ is the corresponding binary label vector.

$$p(X^A) = \underbrace{p(x^N)}_{\substack{\text{modeled} \\ \text{using } p(X^N)}} \cdot \underbrace{p(x^A)}_{\substack{\text{modeled using} \\ p(X^A|X^N, Y^A)}} \cdot \underbrace{p(Y^A)}_{\substack{\text{modeled} \\ \text{using } P(Y^A)}} \tag{1}$$

To learn this factorized generative model, we model the following distributions–

$$\text{Anomaly Label model: } Y^A \sim p(Y^A) \tag{2}$$

$$\text{Sample Occlusion model: } \tilde{X} \sim p(\tilde{X}|X^N, Y^A) \tag{3}$$

$$\text{Foundation model: } \begin{cases} \text{Enc}_\theta : z \sim q_\theta(z|\tilde{X}) \\ \text{Dec}_\psi : X^N \sim p_\psi(X^N|z) \end{cases} \tag{4}$$

$$\begin{matrix} \text{Anomaly} \\ \text{Generation Model:} \end{matrix} \begin{cases} \text{Enc}_{\theta*} : z \sim q_{\theta*}(z|\tilde{X}) \\ \text{Dec}_\phi : X^A \sim p_\phi(X^A|z, Y^A) \end{cases} \tag{5}$$

We start by decomposing the problem into two sub-problems— (i) learning a latent space of the normal sequences; (ii) learning to insert anomalies in latent space. **Stage 1:** We model the first sub-problem using an unconditional VAE (Eq. 4) with an important modification. Instead of learning a vanilla variational encoder $p_\psi(z|X^N)$, we learn $p_\psi(z|\tilde{X})$ where $\tilde{X} \in \mathbb{R}^T$ is the occluded version of a normal sequence $X^N$. Here occlusion refers to removing a set of values from the given TS and replacing with zeros [26]. Instead of occluding values randomly, we occlude values that follow $p(Y^A)$. More specifically, we sample a $Y^A \sim p(Y^A)$ and replace the values in $X^N \sim \mathcal{D}$ with zeros that correspond to timestamps that are labeled as 1 in $Y^A$. This is the *sample occlusion model* in Eq. 3 that gives us $\tilde{X}$. We then train the VAE's encoder and decoder jointly on the paired samples of $(X^N, \tilde{X})$ to maximize the following evidence lower bound (ELBO),

$$\log p(X^N) \geq \mathbb{E}_{q_\theta(z|\tilde{X})}[\log p_\psi(X^N|z)] - KL(q_\theta(z|\tilde{X}) \,||\, p(z)) \tag{6}$$

We use this ELBO objective to write a differentiable loss function below to train $\theta$ and $\psi$ jointly—

$$\mathcal{L}(\theta, \psi) = -KL(q_\theta(z|\tilde{X}_i)||p(z)) + \frac{1}{n} \sum_{i=1}^{n} \log p_\psi(X_i^N|z_i) \tag{7}$$

Both $\theta$ and $\psi$ are implemented using a 2-layer neural network with 32 and 64 units respectively and trained using ADAM optimizer [21] with an initial learning rate of $10^{-3}$. Similar to [22] we assume $p(z)$ as multivariate Gaussian with a diagonal covariance $\mathcal{N}(z; \mu_z, \sigma_z^2 I)$. We are interested in the posterior $q_{\theta*}(z|\tilde{X})$ which learned by above objective can produce a latent representation from an occluded version of a TS sample. This learned posterior will serve as our foundation model to the

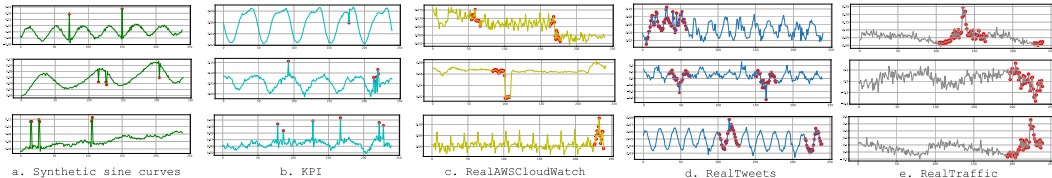

a. Synthetic sine curves     b. KPI     c. RealAWSCloudWatch     d. RealTweets     e. RealTraffic

Figure 2: Labelled anomalous time series generated by `C-GATS` for different datasets showing generality, diversity and consistency of the method.

next stage of training. See pseudo code in Algorithm 1. **Stage 2:** We perform the same occlusion on anomalous samples where we replace the anomalous data points in $X^A$ with zeros to obtain $\tilde{X} \sim p(\tilde{X}|X^A, Y^A)$. Note that post-occlusion the samples obtained from both normal and anomalous sequences belong to the same distribution, i.e $p(\tilde{X}|X^N, Y^A) \approx p(\tilde{X}|X^A, Y^A)$. This allows us to run inference using occluded *anomalous* sequences on a posterior learned on occluded *normal* sequences. We now make use of the anomalous training samples $(X^A, Y^A)$ to train a generator $X^A \sim p_\phi(X^A|z, Y^A)$ conditioned on the labels $Y^A$ where $z$ is obtained from the learned posterior. Note that the posterior model $\theta^*$ is frozen and does not receive gradient updates during this stage of training. We model $\phi$ using a similar 2-layer LSTM decoder architecture as that of $\psi$. The condition vector $Y^A$, which is a binary label corresponding to each timestamp is applied temporally. We found that we must apply this condition at the very last layer of the decoder network to prevent the condition vector from interfering with the reconstruction process of the base signal and restrict it to contribute only in the anomaly insertion process in the high level feature space. This helps in achieving consistency in the generated samples. We show the qualitative and quantitative advantage of this approach in our ablation study in Supp. section 6.7. Training is done in a supervised fashion where $\phi$ is forced to generate an anomalous sequence by inserting anomalies in the latent space where the position and type of anomaly is controlled by the label on which the model is conditioned upon (see Algorithm 2). More specifically, we obtain $\phi^*$ by minimizing the below objective —

$$\min_\phi \sum_{i=1}^{m} - \log p(X_i^A|z_i, Y_i^A; \phi) \tag{8}$$
$$\text{where } z \sim q_{\theta^*}(z|\tilde{X}) \text{ and } \tilde{X} \sim p(\tilde{X}|X^A, Y^A)$$

Once trained, the decoder $\phi^*$ can be used to generate synthetic anomalous TS samples $(X^A, Y^A)$ where $Y^A$ is provided by the user which implicitly controls the attributes of anomaly such as position, duration or type. $z$ is either sampled directly from a normal distribution or from a latent distribution produced by $\theta^*$ when given an occluded version of any normal or anomalous TS from same distribution. The overall architecture of `C-GATS` is shown in Figure 4 in Supplementary.

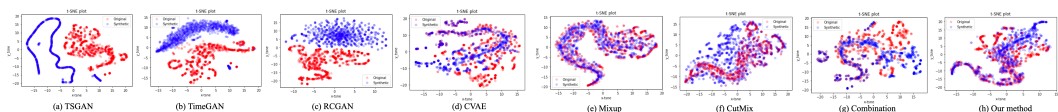

(a) TSGAN    (b) TimeGAN    (c) RCGAN    (d) CVAE    (e) Mixup    (f) CutMix    (g) Combination    (h) Our method

Figure 3: t-SNE visualization on Synthetic Sines dataset. Each plot denotes the visualization of a different TS generation/augmentation method. Red denotes original anomalous TS, and blue denotes the generated anomalous TS.

## 4 Experiments

We compare `C-GATS` with 7 different TS generation/augmentation methods— Mixup [43], CutMix [40], RCGAN [14] and a combination of classical methods like Scaling, Jittering, Permutation and TimeWarp [19] on 6 different public TS datasets using 8 different TS AD algorithms. Following [37], we emphasize that AD is both a quantitative and qualitative study. Hence, we observe the following desiderata as outlined in Section 1— (a) *diversity*; (b) *label-preserving*; (c) *effectiveness*; (d) *model-agnostic*; and (e) *domain-agnostic*. First two are analyzed qualitatively and the next three quantitatively.

**Datasets** We use 6 different datasets to run quantitative assessments of `C-GATS`. These datasets are common in most recent AD papers, and have (a) variety of attributes like noise, periodicity, trend, non-stationarity; (b) different types of anomalies like point anomalies, level-shift, change-point, etc; (c) are collected from different domains like machine telemetry, traffic data. This justifies our claim of domain-agnostic nature of `C-GATS`. Details of each dataset is described in Supp. section 6.2.

## 4.1 Qualitative Analysis

We apply t-SNE [35] on both the generated and real anomalous TS on the Synthetic Sines dataset to visualize the overlap between the two distributions in Figure 3, as in [31]. 3(h) shows a good overlap between the real anomalous TS and that generated by `C-GATS`. Furthermore, we visualize the raw generated samples by `C-GATS` in Figure 2. The red dots denote the inserted anomalies. In Synthetic Sines and KPI datasets, these are point anomalies while in RealAWSCloudWatch they correspond to change points. This provide support to the claim that `C-GATS` is able to generate *diverse* anomalies across dataset from different domains.

Next we qualitatively assess the *label-preserving* nature of `C-GATS`, i.e, if `C-GATS` is able to generate anomalies in position dictated by the conditioned label vector. We use Synthetic Sines dataset where we insert two different types of anomalies— point anomalies and level-shift anomalies randomly in the dataset and train `C-GATS` on it. Once trained, we use the decoder $\phi^*$ to generate anomalous samples $X^A$ by passing as input a noise vector $z$ sampled from a normal distribution along with a conditional vector $Y^A$ as shown in Figure 1. `C-GATS` not only successfully inserts anomalies in the conditioned positions but also attributes the *type* of anomaly to the conditioned label vector, i.e, long anomalous period means level-shift, short anomalous period means point anomalies.

Table 1: Performance of different AD algorithms in different augmentation settings on *Synthetic Sines dataset*

| AD Algorithm | TRTR | | | T(R+S)TR | | | TRTS | | | TSTR | | |
|---|---|---|---|---|---|---|---|---|---|---|---|---|
| | Precision | Recall | F1 | Precision | Recall | F1 | Precision | Recall | F1 | Precision | Recall | F1 |
| **RobustTAD** | 0.76 | 0.73 | 0.74 | 0.76 | 0.75 | 0.75 | 0.74 | 0.70 | 0.71 | 0.77 | 0.75 | 0.75 |
| **SR-CNN** | 0.75 | 0.70 | 0.73 | 0.76 | 0.71 | 0.74 | 0.67 | 0.65 | 0.65 | 0.77 | 0.73 | 0.75 |
| **NCAD** | 0.80 | 0.74 | 0.77 | 0.81 | 0.78 | 0.79 | 0.75 | 0.70 | 0.71 | 0.80 | 0.76 | 0.77 |

Table 2: Performance of different AD algorithms in different augmentation settings on *KPI Dataset*

| AD Algorithm | TRTR | | | T(R+S)TR | | | TRTS | | | TSTR | | |
|---|---|---|---|---|---|---|---|---|---|---|---|---|
| | Precision | Recall | F1 | Precision | Recall | F1 | Precision | Recall | F1 | Precision | Recall | F1 |
| **RobustTAD** | 0.57 | 0.53 | 0.55 | 0.69 | 0.66 | 0.67 | 0.50 | 0.45 | 0.48 | 0.65 | 0.64 | 0.64 |
| **SR-CNN** | 0.55 | 0.51 | 0.54 | 0.61 | 0.55 | 0.58 | 0.47 | 0.43 | 0.44 | 0.60 | 0.55 | 0.57 |
| **NCAD** | 0.60 | 0.68 | 0.64 | 0.70 | 0.69 | 0.69 | 0.51 | 0.54 | 0.52 | 0.65 | 0.73 | 0.68 |

Table 3: Deep TS AD algorithms performance across different datasets

| Algorithm | Dataset | w/o Aug. | Mixup (alpha=0.2) | CutMix | Combination (Scaling, Jitter, Permute, TimeWarp) | RCGAN | `C-GATS` |
|---|---|---|---|---|---|---|---|
| **RobustTAD** | Synthetic Sines | 0.74 ±0.005 | 0.74 ±0.007 | 0.27 ±0.003 | 0.75 ±0.006 | 0.23 ±0.002 | **0.75 ±0.004** |
| | KPI | 0.55 ±0.011 | 0.56 ±0.008 | 0.23 ±0.010 | 0.59 ±0.003 | 0.19 ±0.009 | **0.67 ±0.003** |
| | RealTweets | 0.54 ±0.021 | 0.53 ±0.011 | 0.32 ±0.026 | 0.57 ±0.009 | 0.18 ±0.041 | **0.57 ±0.010** |
| | RealTraffic | 0.67 ±0.026 | 0.68 ±0.029 | 0.37 ±0.020 | 0.69 ±0.012 | 0.27 ±0.071 | **0.73 ±0.015** |
| | RealAWSCloudWatch | 0.34 ±0.262 | 0.34 ±0.089 | 0.06 ±0.344 | **0.36 ±0.218** | 0.04 ±0.181 | 0.34 ±0.192 |
| | ArtificialWithAnomaly | 0.36 ±0.275 | 0.38 ±0.118 | 0.33 ±0.229 | 0.38 ±0.136 | 0.05 ±0.099 | **0.41 ±0.107** |
| **SR-CNN** | Synthetic Sines | 0.73 ±0.009 | 0.72 ±0.002 | 0.18 ±0.009 | 0.74 ±0.001 | 0.16 ±0.010 | **0.74 ±0.003** |
| | KPI | 0.54 ±0.008 | 0.55 ±0.005 | 0.19 ±0.009 | 0.56 ±0.003 | 0.17 ±0.011 | **0.58 ±0.007** |
| | RealTweets | 0.46 ±0.027 | 0.45 ±0.019 | 0.24 ±0.033 | 0.51 ±0.009 | 0.19 ±0.008 | **0.53 ±0.009** |
| | RealTraffic | 0.64 ±0.021 | 0.65 ±0.017 | 0.33 ±0.019 | 0.65 ±0.028 | 0.26 ±0.035 | **0.66 ±0.020** |
| | RealAWSCloudWatch | 0.25 ±0.319 | 0.25 ±0.283 | 0.04 ±0.081 | **0.28 ±0.193** | 0.01 ±0.198 | 0.27 ±0.199 |
| | ArtificialWithAnomaly | 0.28 ±0.019 | 0.30 ±0.174 | 0.24 ±0.111 | 0.32 ±0.081 | 0.12 ±0.059 | **0.34 ±0.011** |
| **NCAD** | Synthetic Sines | 0.77 ±0.008 | 0.76 ±0.001 | 0.30 ±0.003 | 0.78 ±0.019 | 0.27 ±0.071 | **0.79 ±0.001** |
| | KPI | 0.64 ±0.006 | 0.66 ±0.003 | 0.24 ±0.009 | 0.67 ±0.010 | 0.21 ±0.004 | **0.69 ±0.001** |
| | RealTweets | 0.57 ±0.019 | 0.55 ±0.014 | 0.32 ±0.026 | 0.58 ±0.010 | 0.20 ±0.085 | **0.61 ±0.009** |
| | RealTraffic | 0.69 ±0.029 | 0.69 ±0.033 | 0.43 ±0.098 | 0.71 ±0.051 | 0.28 ±0.171 | **0.71 ±0.019** |
| | RealAWSCloudWatch | 0.35 ±0.219 | 0.34 ±0.118 | 0.11 ±0.092 | **0.37 ±0.071** | 0.07 ±0.215 | 0.35 ±0.111 |
| | ArtificialWithAnomaly | 0.39 ±0.101 | 0.42 ±0.213 | 0.36 ±0.128 | 0.42 ±0.098 | 0.13 ±0.023 | **0.45 ±0.091** |

## 4.2 Quantitative Analysis

**Evaluation Frameworks** We select 3 supervised NN-based, and 5 baseline TS anomaly detection (AD) algorithms to benchmark the improvement in anomaly detection quality using `C-GATS`. Details about these baseline algorithms are provided in Supp. section 6.6. We pick 7 different DA techniques to compare against C-GATS and details are in section 6.5. We adopt 4 evaluation frameworks [14], namely– TRTR (Train on real, Test on real), TSTR (Train on synthetic, Test on real), TRTS (Train on real, Test on synthetic) and T(R+S)TR (Train on real+synthetic, Test on real). We split each of the 6 datasets into a 80:20 train:test ratio. We use the $80\%$ train set to train `C-GATS` and generate equal no. of synthetic samples as there are in the train set. More details about these frameworks are provided in Supp. section 6.3. For results, we report the average point-based precision, recall and F1 scores and justify the choice of these metrics in Supp. section 6.4.

**Benchmark Results** Tables 1-2 along with tables 5-8 in the supplementary show the performance of three neural-net based AD algorithms in different evaluation frameworks for each of 6 datasets. **(1)** We compare the TRTR and T(R+S)TR frameworks to see if augmenting the training data with `C-GATS` generated anomalous samples further improves the AD performance? On the simulated dataset where the training set is large and diverse (Synthetic Sines) the improvement is minimal ($< 2\%$) but on a more realistic dataset (KPI) the improvement is higher, $\sim 18\%$ increase in F1 score. **(2)** TSTR results help identify if generated samples actually lead to a useful AD model. We see that on both simulated and real-world datasets, the performance is equivalent and in some cases better than TRTR case. This indicates that the generated samples are not only realistic but also consistent with their corresponding ground truth labels leading to a comparable performance on real test sets. **(3)** In TRTS setting we see that the performance of AD models decline as compared to TRTR. This means if trained on real-world samples and tested on generated anomalous samples the detection quality declines. One could argue that this could mean the generates samples are corrupt/incorrect. But since TSTR leads to a higher performance than TRTR, we can safely conclude that the generated samples are not only good quality and correctly labelled but also much more diverse than the real dataset. **(4)** Table 3 shows the mean and standard deviation of F1 scores across 5 runs for each setting. The column of w/o Aug. simply represents the numbers from TRTR framework while all the other augmentation methods are evaluted in T(R+S)TR framework. We observe that— **(a)** Classic vision based augmentation strategies like CutMix [40] or Mixup [43] are not as effective when compared to an ensemble of traditional TS augmentation methods like scaling, jittering, permutation or TimeWarp on anomalous data. This is because the former approaches sometimes lead to corrupt anomalous samples with incorrect ground truth labels (see Figure 6 in Supp.) causing poor AD performance. **(b)** Complex GAN-based method like RCGAN [14] lead to unstable training and fail to learn sparse anomalous signal in the long complex TS. This leads to generation of unrealistic anomalous sequences and hence poor AD performance. **(c)** `C-GATS` outperforms all these augmentation methods for anomalous TS data on 5 out 6 datasets. However, when training set is small (eg: RealAWSCloudWatch dataset), `C-GATS` fails to learn good realistic patterns and hence in those cases an ensemble of classical DA methods outperforms `C-GATS`. **(d)** The algorithms Case 1-3, One-liner A and B serve as robust baselines to avoid creating an illusion of improvement in detection quality when using the DA method, as these baselines remain largely unaffected by augmentation strategies. We notice that some of these baselines, e.g. One-liner A and B have comparable, and in some cases, even better performance than some of the deep AD algorithms with augmentations (See Table 9 in Supp.). This echoes the claims made in [37] that deep learning might not always be the right solution to problem in TS AD domain. We further add that complex DA strategies do not boost the detection quality significantly in cases when the anomalies are trivial to detect.

## 5 Conclusion & Future Work

We present a simple semi-supervised approach to generate synthetic anomalous TS data with labels, and demonstrate that the generated samples exhibit properties similar to the anomalies in real dataset. What is unique is that our model is able to learn and embed invariances present in the real data into synthetic data without any extra supervision, leading to generation of more diverse anomalous samples as compared to traditional DA approaches. In Supp section 6.8 we discuss how this idea can be further extended to achieve not just diverse but more complex set of synthetic anomalies.

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

# 6 Supplementary Material

## 6.1 Related Work

### 6.1.1 Generating synthetic data using known invariances

Making deep learning models robust when they are trained on limited data is critical. One approach is to encode known invariances into the model directly [18]. For example a classifier trained to detect cat images should be invariant to rotated, cropped or blurred images of cat. However, it is non-trivial to encode all invariances directly into the model. A simpler way is to encode those invariances in the training data by generating additional data using transformations. Traditional DA techniques used for this purpose employ a limited set of invariances that are static, known a priori, and easy to implement [13, 43, 40, 19]. Although considered best practices for a long time [30], this class of methods are shown to have limitations such as incapability of exploring a large invariance space [1] and requiring domain knowledge to design augmentation schemes [18].

### 6.1.2 Generating synthetic data by learning invariances

Recently a lot of work has been done in vision domain to infer invariances directly from training data [28, 9, 10, 1, 18] using generative models like GANs/VAEs that have been successfully adapted to TS domain [8, 39, 14, 34, 15], to generate synthetic TS samples that show effectiveness primarily on the task of TS classification. [39] supplements a GAN based generative model with a supervised autoregressive objective in the latent space and shows that it leads to more realistic TS that improve performance on a classification task. [8] use a GAN based approach and learn a modal-agnostic generative framework by jointly training a classifier in the latent space. Although the intuition behind GANs is quite elegant as exhibited in these papers, they are difficult to train and often experience mode collapse (which prevents from generalizing), vanishing gradients, and/or unstable updates. In [11] authors use a deterministic approach and train a denoising auto encoder [36] to generate financial TS data. Although impressive, there is no evidence that these complex GAN-based or compression-based auto encoding methods would work equally well on anomalous TS where its much harder to learn to generate sparse signals like point anomalies and level-shifts, in the synthetic data. In [6, 7] authors show anomalous TS can synthetically be generated by sampling from a learned latent space. Although impressive, there are two key limitations- (a) they still use a limited set of known invariances (eg: jittering, scaling, permutation) to handle class imbalance to train a VAE; (b) they pose AD task as a classification task to alleviate the problem of dealing with temporal labels which limits this approach to be of use in a traditional TS AD setting [37].

**Conditional Generative Models.** Explicitly conditioning the generative models on class labels has shown to learn a sharper and class-dependent data distribution. This idea has been adopted to modify both VAEs [32] and GANs [31, 27, 14] for TS generation. In [31] authors train two Wasserstein GANs [2] in a sequential manner where the spectograms generated from first WGAN are used to condition the second WGAN, which is trained to generate synthetic TS data. In [27] authors apply conditioning on timestamp information to handle irregular sampling while [14] applies conditioning on class labels in a temporal fashion to both the generator and discriminator to learn a class-dependent generative model.

**VAEs for data imputation.** Another interesting line of work involving VAEs for TS generation is data imputation. In [26] authors show how VAEs can be used to fill missing data in a high-dimensional heterogenous setting which was extend to TS by [16] by building a sequential latent variable model and using a prior that exploits temporal dynamics in latent space. [42] is another recent work where VAEs are proven to be useful for imputing missing TS data.

Inspired by these recent successes of VAEs we show how `C-GATS` adopts a factorized training procedure and uses ideas from data imputation to train an unconditional VAE on just the normal samples that acts as a foundation model [4] which is then used to fine-tune a conditional generative model that learns to generate synthetic anomalous TS with labels and addresses the limitations of previous approaches.

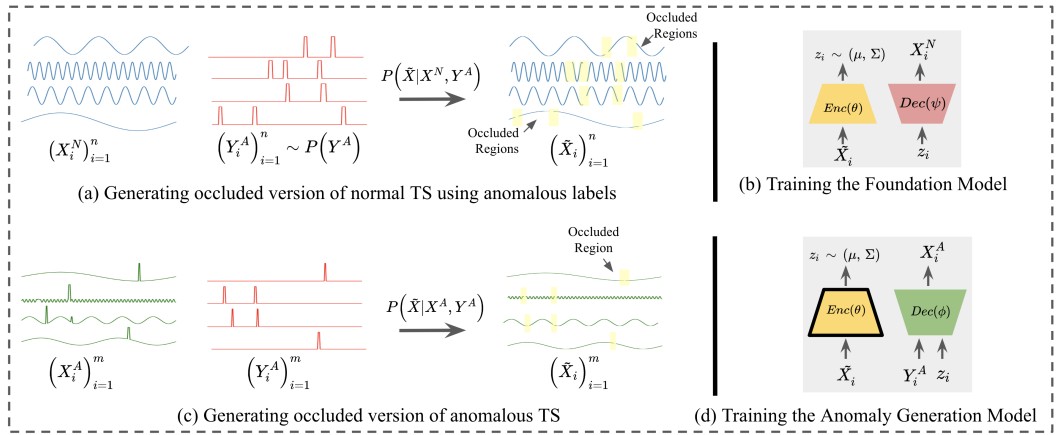

Figure 4: `C-GATS` Architecture. (a) Step 1: *sample occlusion model* is used to obtain $\tilde{X}$ from normal TS; (b) Step 2: Paired samples $(\tilde{X}, X^N)$ are then used to train the foundation model $\theta$; (c) Step 3: *sample occlusion model* is used to obtain $\tilde{X}$ from anomalous TS; (d) Step 4: Anomaly generation model $\phi$ is trained in a conditional manner using $(\tilde{X}_i, X_i^A, Y_i^A)_{i=1}^m$. Solid lines around $\theta$ in (d) denote frozen weights.

## 6.2 Dataset Description

**Synthetic Sines**– We simulate univariate sinusoidal sequences of different frequencies $\eta$ and phases $\theta$, providing continuous-valued periodic samples each of fixed length, $T = 240$ timestamps; $x_i(t) = sin(2\pi\eta t + \theta)$, where $\eta \sim \mathcal{U}[0, 1]$ and $\theta \sim \mathcal{U}[-\pi, \pi]$. We simulate a total of 64000 TS. Then we insert point anomalies in the data. We sample 3 attributes randomly at uniform– (1) whether to insert an anomaly or not $s \sim \{1, 0\}$; (2) the position where to insert the anomaly, $p \sim range(240)$; (3) how long will the anomaly be, $l \sim range(10)$; (4) direction of anomaly, i.e, either positive of negative spike $d \sim \{+1, -1\}$. Thus, for sequence $x_i$ in simulated data, we corrupt it by inserting anomaly using the process- $x_i^A = f(x_i; s, p, l, d)$.

**KPI dataset**[1]– A univariate TS dataset consisting of KPI curves from different internet companies in 1 minute interval. We use a sliding window of 240 to downsample the long TS in the data and obtain a fixed size datasets of 30000 TS each of length 240 timesteps.

**NAB**[2]– A public anomaly detection benchmark dataset [23] that contain streaming data from different domains. We select 4 different datasets from this benchmark- **(a) RealTweets**, **(b) RealTraffic**, **(c) RealAWSCloudWatch**, **(d) ArtificialWithAnomaly**. Each dataset was resampled to sequences of fixed length. Statistics of each dataset are provided in Table 4.

Table 4: Statistics of datasets

| Dataset | Total Series | Total Points | Anomaly Points |
|---|---|---|---|
| **Synthetic Sines** | 64000 | 15360000 | 171071 |
| **KPI** | 30000 | 7200000 | 328527 |
| **RealTweets** | 4000 | 960000 | 159740 |
| **RealTraffic** | 1200 | 288000 | 14364 |
| **RealAWSCloudWatch** | 121 | 29040 | 3070 |
| **ArtificialWithAnomaly** | 180 | 43200 | 5040 |

## 6.3 Evaluation Mechanisms

- **TRTR**: Train on the $80\%$ real train set and evaluate on the $20\%$ held-out real test set.

---

[1]http://iops.ai/competition_detail/?competition_id=5&flag=1

[2]https://github.com/numenta/NAB/tree/master/data

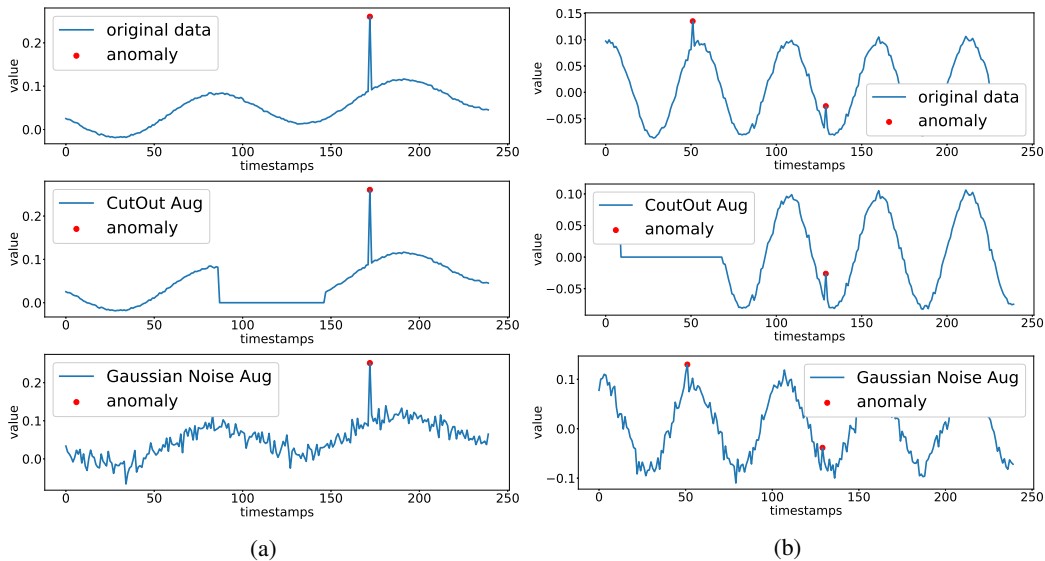

Figure 5: We augment 2 different sine curves with 2 popular DA methods from vision domain that have been applied to TS [38]- first, CutOut [13] where we randomly replace a small section of TS with 0; second, we add random noise sampled from a normal distribution. In both cases DA fails– either Standard DA techniques either introduce new anomalous behavior in the data or corrupt the anomalous timestamps leading to incorrect ground-truth labels.

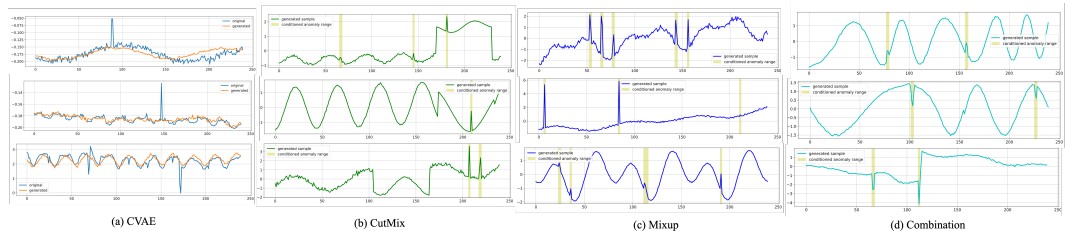

Figure 6: Samples generated by four different augmentation methods. Column (a) shows 3 different samples generated by a CVAE; Blue curve represents the input sample and orange curve represents the generated sample. Column (b), (c) and (d) show samples generated by CutMix [40], Mixup [43] and Combination (i.e jittering, scaling, permute and timewarp) augmentation methods.

- **T(R+S)TR**: Train on the combined data (i.e 80% train set and all of the generated synthetic data) and test on the held-out 20% real test set.

- **TSTR**: Train on the all of the generated synthetic data and test on the held-out $20\%$ real test set.

- **TRTS**: Train on all the real data $(80\% + 20\%)$ and test on a $20\%$ random split of synthetic data.

## 6.4 Evaluation Metrics

We observed that most of these AD methods [17, 29, 5] report performance scores using either a point-adjusted scoring function [3] or a relaxed version of F1 [17], which leads to an overestimation of detection performance [20]. We therefore use point-based evaluation metrics as recommended by [20] and also use the baselines by [37] and [20]. We report the average point-based precision, recall and F1 scores for each of the AD evaluation experiment performed. A series of AD experiments are run on each of the 6 datasets where we evaluate all the 8 different AD algorithms with 6 different DA strategies. Each experiment was run 5 times with different splits and we report the mean and standard deviation of the F1 score in Table 3, 9, 10 and 11.

### 6.5 DA Baselines

We pick 7 different DA techniques to compare against `C-GATS`. We also include an eighth baseline which is simply not using any augmentation method. Mixup [43] creates new training examples out of original samples by using a convex combinations of the features and their labels (controlled by $\alpha$) resulting into plausible new TS (e.g. see Figure 6.c). We choose $\alpha = 0.2$ for our experiments as it yielded the best results across different values of $\alpha$. CutMix [40] is another strategy popular in vision domain which involves randomly cutting a patch from one sequence and pasting it into another sequence. The new sequences generated by this method are not very ideal for anomaly detection task as abruptly changing patterns in a sequence could introduce new anomalies that labels won't account for (see Figure 6.b). As third baseline we pick RCGAN [14] as it is a neural-network based approach that applies conditionals in a temporal fashion like `C-GATS`. Out fourth baseline is a combination of traditional TS augmentation methods [3] including Scaling, Jittering, Permutation and TimeWarping. In this method, for each sequence in a given dataset, we randomly sample one DA strategy from the mix and apply it to the sequence. (e.g. see Figure 6.d).

### 6.6 AD Baselines

We pick 3 supervised neural-network based AD algorithms– RobustTAD [17], SR-CNN [29] and NCAD [5] as each claim to utilize DA techniques for better detection performance making them an ideal choice for our experiments. Apart from these, we select 5 other baselines that are much simpler but have been shown to be effective on the TS AD benchmark datasets [37, 20]. We refer to them as One-liner A, One-liner B, Case 1, 2 and 3. **Case 1** [20] baseline randomly assigns an anomaly score for every timestamp in a given sequence, i.e $\mathcal{A}(x_t) \sim \mathcal{U}(0, 1)$. **Case 2** [20] baseline assigns a score proportional to the value at each timestamp, i.e $\mathcal{A}(x_t) = ||x_t - \eta||_2$ where $\eta = 0$. **Case 3** [20] baseline is same as Case 2 but $\eta$ is obtained from an untrained 2-layer LSTM neural network whose parameters are fixed after being initialized from a Gaussian distribution $\mathcal{N}(0, 0.02)$. The scores are then converted into anomaly labels using a threshold $0 \leq \delta \leq 1$. We do a grid search for the value of $\delta$ and report the metrics for the best value. Another set of baselines obtained from [37] are one-liners. There are 2 main types of one-liners proposed in [37] one with $abs$ and without. We test both the categories and pick the one that obtain best performance. We pick **One-liner A** as —

```
abs(diff(TS)) > b
```

and **One-liner B** as —

```
abs(diff(TS)) > movmean(diff(TS), k)
             + c * movstd(diff(TS), k) + b
```

As recommended by [37], we adopt a similar brute-force strategy to compute individual **k**, **c**, **b** for all the datasets.

### 6.7 Ablation Study

To better understand the advantage brought by different components of our method, we perform an ablation study on 6 different datasets and 3 different AD algorithms. We consider variations of the framework by (a) varying the training procedure; and (b) altering the model architecture and train the model in each configuration thrice. We report the average performance metrics of these runs.

#### 6.7.1 Dual-step training strategy

In this experiment we remove the training phase-I from `C-GATS` and simply train a standard CVAE with temporal conditionals with anomalous samples. Table 10 summarizes the performance of `C-GATS` in these two different settings. Our study shows that the use of dual-training step contributes to decoupling complex processes of representation learning and anomaly insertion which leads to generation of more controlled and sharper anomalies and contributes to additional performance gain. We further demonstrate this in Figure 7 where the left hand column represents sampled generated

---

[3]https://tsaug.readthedocs.io/en/stable/

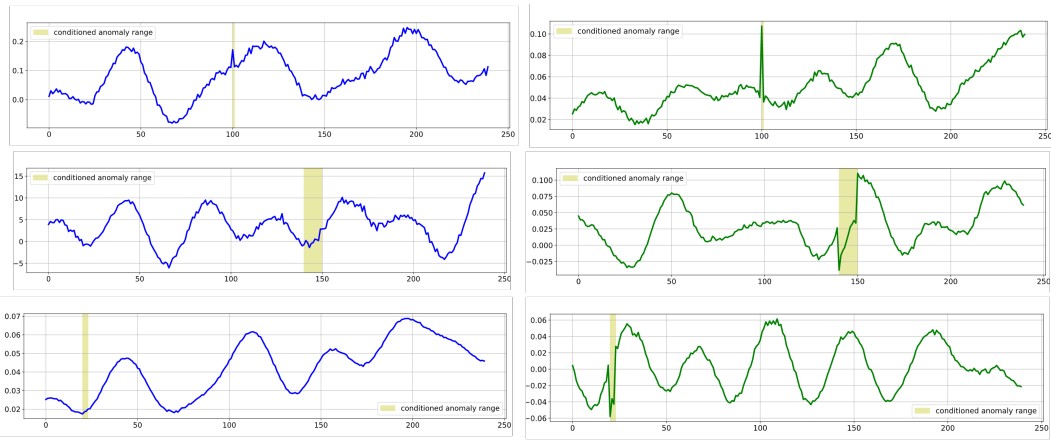

Figure 7: For a fixed $z$ and $Y^A$ in each row; Blue: samples generated by C-GATS in 1-phase training; Green: samples generated by C-GATS in a 2-phase training.

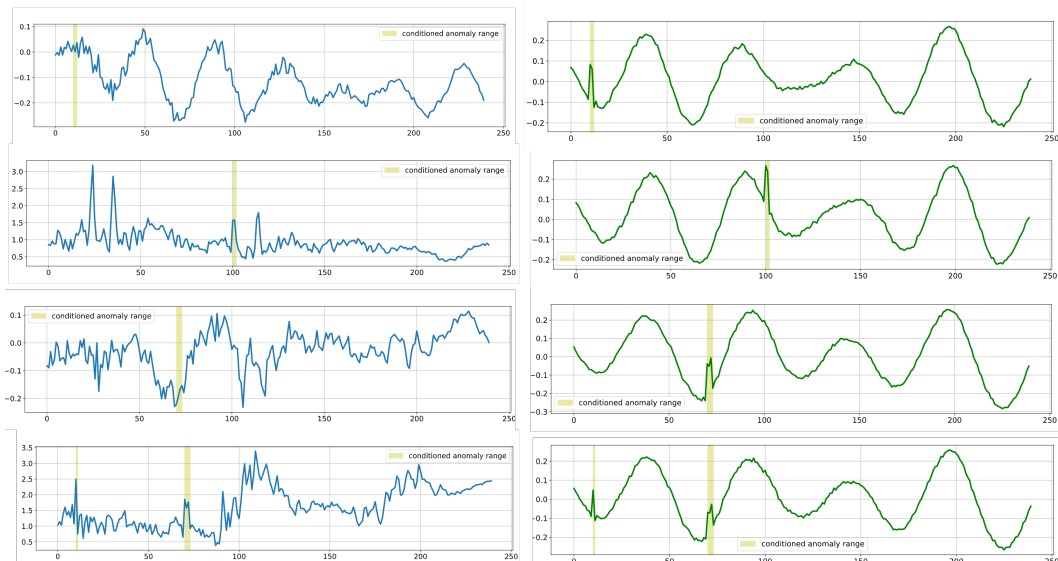

Figure 8: For a fix latent vector $z$, we vary $Y^A$ in each row. Blue plots: represent samples generated by C-GATS when temporal conditioning is applied at the very first layer of the generator. Green plots: samples generated when conditioning is applied at the last layer of generator.

via one-step training procedure and right-hand column represents the dual-step training of C-GATS on KPI dataset. For each row, we use a fixed noise vector $z$ sampled from a normal distribution along with a fixed condition vector $Y^A$. The samples generated by on-step training fail to learn sharp distinguishable anomalies in data which can be seen in the samples generated by the two-step training strategy.

### 6.7.2 Positioning of Conditionals

We study the effectiveness of our proposed approach of applying temporal conditionals for anomaly generation. By varying the position of the temporal conditionals we study its impact on the generation process. Changing the position of applying conditionals from final-most layer to the very first layer of the anomaly generator leads to a decline in AD performances by upto 12% in some cases, see Table 11. Neural networks are known to learn more fundamental and primitive features at the initial layers while more advanced and developed ones at the later layers [41]. Similar phenomena is observed when we assess the quality of generated anomalous samples in Figure 8. The study reveals that applying conditionals at the first layer of the generator interferes with the basic reconstruction of TS

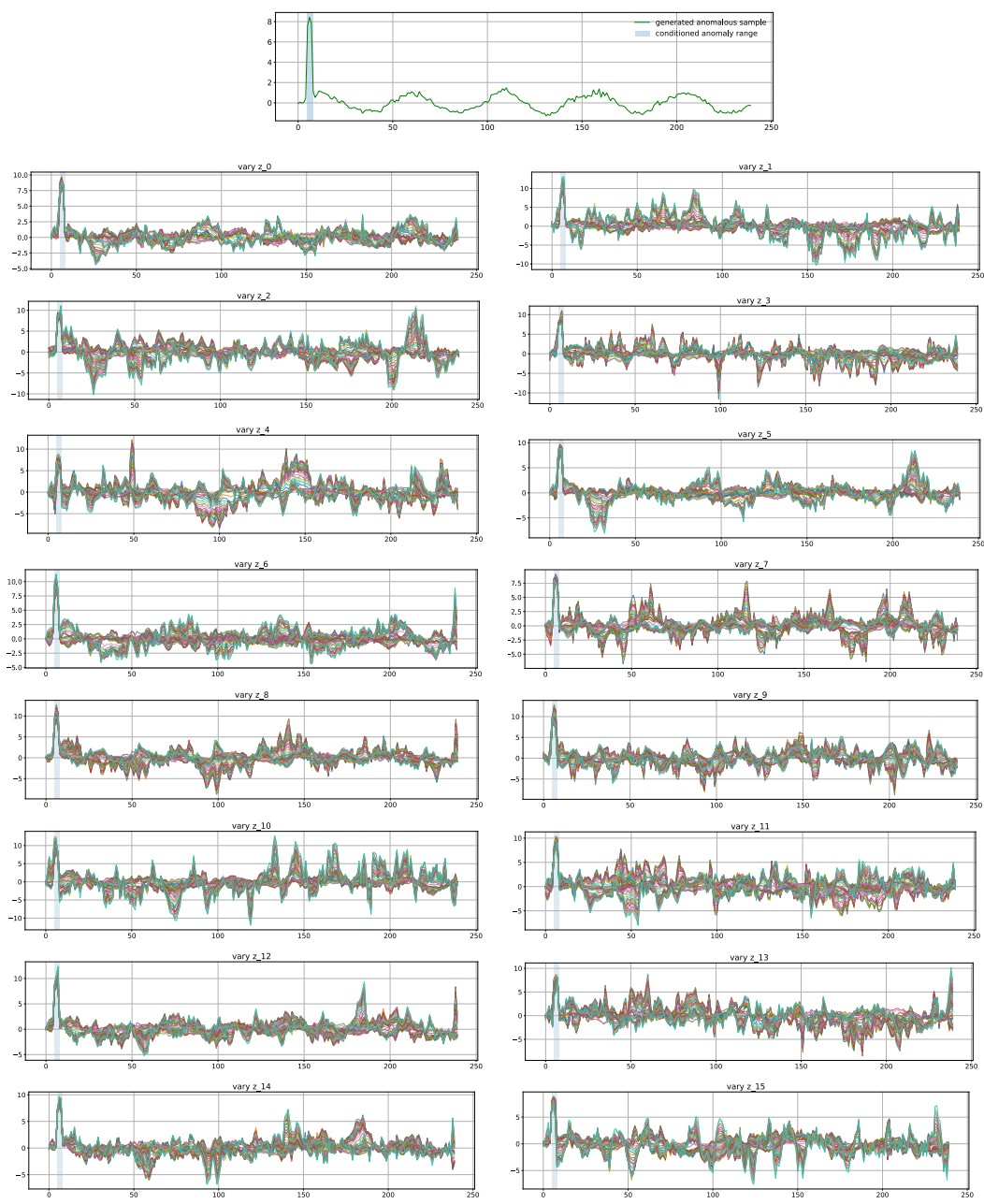

Figure 9: Top center: For a given conditional vector $Y^A$ and a latent vector $z$ sampled from a standard Gaussian, we generate an anomalous sequence. Then we vary $z$ across each of its 16 dimensions one by one keeping the rest constant. For a given conditional vector and a latent $z$ `C-GATS` can generate multiple reasonable and interesting synthetic anomalous signals.

signal and leads to ambiguous and inconsistent anomalous samples. Whereas when applied at the final-most layer, the generated samples are consistent and better in quality and hence contribute to better AD performance.

### 6.7.3 Diversity in generation of anomalies across a fixed input

Here we assess the generative power of `C-GATS` qualitatively in Figure 9. For a fixed condition vector $Y^A$ we take a random noise $z$ which is a 16-dimensional vector and generate an anomalous

---

**Algorithm 1** Training the foundation model

---

**Input**: an anomaly label model $p(Y^A)$; a sample occlusion model $p(\tilde{X}|X^N, Y^A)$; set of normal sequences $\{X_i^N\}_{i=1}^n$; batch size $B$.
**Parameter**: $\theta$ and $\psi$ initialized randomly.
**Output**: A learned posterior model $p(z|\tilde{X}; \theta^*)$.

1: **while** SGD *not converged* **do**
2:     Sample a batch of $B$ normal sequences $(X_i^N)_{i=1}^B$;
3:     Initialize an empty buffer $\mathcal{M}$ of size $B$;
4:     **for** $i = 1, \ldots, B$ **do**
5:         Sample $Y^A$ from *label model* $Y^A \sim P(Y^A)$;
6:         Feed $(X_i^N, Y^A)$ to *sample occlusion model* and obtain $\tilde{X}_i \sim p(\tilde{X}|X_i^N, Y^A)$;
7:         Collect samples $\mathcal{M} \leftarrow \mathcal{M} \cup \{(\tilde{X}, X^N)_i\}$;
8:     **end for**
9:     $\epsilon \sim p(\epsilon)$; (Random noise for every datapoint in $\mathcal{M}$)
10:     Compute $\mathcal{L}_{\theta,\psi}(\mathcal{M}, \epsilon)$ via Eq.(7) and its corresponding gradients $\nabla_{\theta,\psi}\mathcal{L}_{\theta,\psi}(\mathcal{M}, \epsilon)$;
11:     Update $\theta$ and $\psi$ using SGD optimizer;
12: **end while**

---

**Algorithm 2** Training the anomaly generation model

---

**Input**: a trained variational encoder $p(z|\tilde{X}; \theta^*)$; a sample occlusion model $p(\tilde{X}|X^A, Y^A)$; set of anomalous sequences and labels $\{X_i^A, Y_i^A\}_{i=1}^m$; batch size $B$.
**Parameter**: $\phi$ initialized randomly.
**Output**: A learned generative model $p(X^A|z, Y^A; \phi^*)$.

1: **while** SGD *not converged* **do**
2:     Sample a batch of $B$ anomalous sequences pairs $(X_i^A, Y_i^A)_{i=1}^B$;
3:     Initialize an empty buffer $\mathcal{M}$ of size $B$;
4:     **for** $i = 1, \ldots, B$ **do**
5:         Feed $(X_i^A, Y_i^A)$ to *sample occlusion model* and obtain $\tilde{X}_i \sim p(\tilde{X}|X_i^A, Y^A)$;
6:         Feed $\tilde{X}_i$ to the trained encoder and obtain the latent representation $z_i \sim p(z|\tilde{X}_i; \theta^*)$
7:         Collect samples $\mathcal{M} \leftarrow \mathcal{M} \cup \{(z_i, X_i^A, Y_i^A)\}$;
8:     **end for**
9:     Compute $\mathcal{L}_\phi(\mathcal{M})$ (8) and gradients $\nabla_\phi\mathcal{L}_\phi(\mathcal{M})$;
10:     Update $\phi$ using SGD optimizer;
11: **end while**

---

sample as shown in the top-center of the figure. We then vary each of 16 dimensions of $z$ one-by-one between values $(-1, 1)$ keeping the rest constant. The figure shows that on varying each dimension of $z$, we control different attributes of the generated sample such as frequency, noise, etc. while retaining the anomaly in the same desired location. This displays the generative power of `C-GATS` in producing a diverse range of synthetic anomalous samples.

## 6.8   Future Work

In the future, we plan to further improve `C-GATS`'s generation quality by forcing it to generate not just realistic but hard examples that can further improve the quality of AD models. Similar ideas have been used in approaches like MODALS [8] where sampling in latent space is advised by a classifier to pick samples that can potentially fool the classifier. We believe adapting a similar strategy in `C-GATS` where we use a group of different anomaly detectors to aid the sampling process in latent space could force `C-GATS` to generate samples in feature space that are not only anomalous but can simultaneously fool these detectors, and hence improve downstream AD methods substantially. Another area of improvement is to model the latent distribution using a prior that captures temporal dynamics like [16].

Table 5: Performance of different AD algorithms in different augmentation settings on *RealTweets Dataset*

| AD Algorithm | TRTR | | | T(R+S)TR | | | TRTS | | | TSTR | | |
|---|---|---|---|---|---|---|---|---|---|---|---|---|
| | Precision | Recall | F1 | Precision | Recall | F1 | Precision | Recall | F1 | Precision | Recall | F1 |
| RobustTAD | 0.53 | 0.57 | 0.54 | 0.59 | 0.59 | 0.57 | 0.47 | 0.54 | 0.49 | 0.57 | 0.57 | 0.56 |
| SR-CNN | 0.49 | 0.51 | 0.46 | 0.55 | 0.54 | 0.53 | 0.45 | 0.50 | 0.47 | 0.52 | 0.53 | 0.50 |
| NCAD | 0.55 | 0.60 | 0.57 | 0.61 | 0.64 | 0.61 | 0.49 | 0.55 | 0.53 | 0.58 | 0.63 | 0.59 |

Table 6: Performance of different AD algorithms in different augmentation settings on *RealTraffic Dataset*

| AD Algorithm | TRTR | | | T(R+S)TR | | | TRTS | | | TSTR | | |
|---|---|---|---|---|---|---|---|---|---|---|---|---|
| | Precision | Recall | F1 | Precision | Recall | F1 | Precision | Recall | F1 | Precision | Recall | F1 |
| RobustTAD | 0.74 | 0.65 | 0.67 | 0.78 | 0.70 | 0.73 | 0.64 | 0.63 | 0.62 | 0.76 | 0.69 | 0.69 |
| SR-CNN | 0.69 | 0.60 | 0.64 | 0.71 | 0.63 | 0.66 | 0.68 | 0.55 | 0.58 | 0.70 | 0.61 | 0.65 |
| NCAD | 0.75 | 0.66 | 0.69 | 0.78 | 0.69 | 0.71 | 0.75 | 0.65 | 0.67 | 0.77 | 0.67 | 0.69 |

Table 7: Performance of different AD algorithms in different augmentation settings on *RealAWSCloud-Watch Dataset*

| AD Algorithm | TRTR | | | T(R+S)TR | | | TRTS | | | TSTR | | |
|---|---|---|---|---|---|---|---|---|---|---|---|---|
| | Precision | Recall | F1 | Precision | Recall | F1 | Precision | Recall | F1 | Precision | Recall | F1 |
| RobustTAD | 0.30 | 0.48 | 0.34 | 0.31 | 0.48 | 0.34 | 0.25 | 0.42 | 0.29 | 0.30 | 0.48 | 0.34 |
| SR-CNN | 0.27 | 0.42 | 0.25 | 0.29 | 0.43 | 0.27 | 0.22 | 0.37 | 0.25 | 0.26 | 0.42 | 0.25 |
| NCAD | 0.29 | 0.49 | 0.35 | 0.31 | 0.49 | 0.35 | 0.24 | 0.45 | 0.26 | 0.28 | 0.49 | 0.35 |

Table 8: Performance of different AD algorithms in different augmentation settings on *ArtificialWith-Anomaly Dataset*

| AD Algorithm | TRTR | | | T(R+S)TR | | | TRTS | | | TSTR | | |
|---|---|---|---|---|---|---|---|---|---|---|---|---|
| | Precision | Recall | F1 | Precision | Recall | F1 | Precision | Recall | F1 | Precision | Recall | F1 |
| RobustTAD | 0.44 | 0.32 | 0.36 | 0.50 | 0.37 | 0.41 | 0.41 | 0.30 | 0.34 | 0.47 | 0.35 | 0.37 |
| SR-CNN | 0.39 | 0.17 | 0.28 | 0.43 | 0.26 | 0.34 | 0.37 | 0.15 | 0.25 | 0.41 | 0.21 | 0.29 |
| NCAD | 0.45 | 0.36 | 0.39 | 0.50 | 0.42 | 0.45 | 0.43 | 0.35 | 0.37 | 0.48 | 0.40 | 0.41 |

Table 9: Baseline TS AD algorithms performances across different datasets

| Algorithm | Dataset | w/o Aug. | Mixup (alpha=0.2) | CutMix | Combination (Scaling, Jitter, Permute, TimeWarp) | RCGAN | C-GATS |
|---|---|---|---|---|---|---|---|
| One-liner A | Synthetic Sines | 0.70 ±0.000 | 0.70 ±0.000 | 0.70 ±0.000 | 0.70 ±0.000 | 0.70 ±0.000 | 0.70 ±0.000 |
| | KPI | 0.61 ±0.000 | 0.61 ±0.000 | 0.61 ±0.000 | 0.61 ±0.000 | 0.61 ±0.000 | 0.61 ±0.000 |
| | RealTweets | 0.49 ±0.000 | 0.49 ±0.000 | 0.49 ±0.000 | 0.49 ±0.000 | 0.49 ±0.000 | 0.49 ±0.000 |
| | RealTraffic | 0.51 ±0.000 | 0.51 ±0.000 | 0.51 ±0.000 | 0.51 ±0.000 | 0.51 ±0.000 | 0.51 ±0.000 |
| | RealAWSCloudWatch | 0.38 ±0.000 | 0.38 ±0.000 | 0.38 ±0.000 | 0.38 ±0.000 | 0.38 ±0.000 | 0.38 ±0.000 |
| | ArtificialWithAnomaly | 0.19 ±0.000 | 0.19 ±0.000 | 0.19 ±0.000 | 0.19 ±0.000 | 0.19 ±0.000 | 0.19 ±0.000 |
| One-liner B | Synthetic Sines | 0.69 ±0.000 | 0.69 ±0.000 | 0.69 ±0.000 | 0.69 ±0.000 | 0.69 ±0.000 | 0.69 ±0.000 |
| | KPI | 0.56 ±0.000 | 0.56 ±0.000 | 0.56 ±0.000 | 0.56 ±0.000 | 0.56 ±0.000 | 0.56 ±0.000 |
| | RealTweets | 0.49 ±0.000 | 0.49 ±0.000 | 0.49 ±0.000 | 0.49 ±0.000 | 0.49 ±0.000 | 0.49 ±0.000 |
| | RealTraffic | 0.52 ±0.000 | 0.52 ±0.000 | 0.52 ±0.000 | 0.52 ±0.000 | 0.52 ±0.000 | 0.52 ±0.000 |
| | RealAWSCloudWatch | 0.36 ±0.000 | 0.36 ±0.000 | 0.36 ±0.000 | 0.36 ±0.000 | 0.36 ±0.000 | 0.36 ±0.000 |
| | ArtificialWithAnomaly | 0.18 ±0.000 | 0.18 ±0.000 | 0.18 ±0.000 | 0.18 ±0.000 | 0.18 ±0.000 | 0.18 ±0.000 |
| Case 1 | Synthetic Sines | 0.09 ±0.004 | 0.09 ±0.008 | 0.09 ±0.003 | 0.09 ±0.002 | 0.09 ±0.003 | 0.09 ±0.001 |
| | KPI | 0.08 ±0.006 | 0.09 ±0.003 | 0.08 ±0.009 | 0.08 ±0.010 | 0.09 ±0.004 | 0.08 ±0.001 |
| | RealTweets | 0.12 ±0.009 | 0.11 ±0.004 | 0.12 ±0.001 | 0.12 ±0.010 | 0.11 ±0.008 | 0.12 ±0.003 |
| | RealTraffic | 0.07 ±0.002 | 0.06 ±0.010 | 0.07 ±0.005 | 0.07 ±0.008 | 0.06 ±0.001 | 0.07 ±0.006 |
| | RealAWSCloudWatch | 0.12 ±0.002 | 0.12 ±0.007 | 0.11 ±0.001 | 0.11 ±0.011 | 0.12 ±0.005 | 0.11 ±0.004 |
| | ArtificialWithAnomaly | 0.14 ±0.000 | 0.13 ±0.003 | 0.14 ±0.003 | 0.14 ±0.001 | 0.14 ±0.003 | 0.14 ±0.001 |
| Case 2 | Synthetic Sines | 0.43 ±0.000 | 0.43 ±0.000 | 0.43 ±0.000 | 0.43 ±0.000 | 0.43 ±0.000 | 0.43 ±0.000 |
| | KPI | 0.33 ±0.000 | 0.33 ±0.000 | 0.33 ±0.000 | 0.33 ±0.000 | 0.33 ±0.000 | 0.33 ±0.000 |
| | RealTweets | 0.21 ±0.000 | 0.21 ±0.000 | 0.21 ±0.000 | 0.21 ±0.000 | 0.21 ±0.000 | 0.21 ±0.000 |
| | RealTraffic | 0.14 ±0.000 | 0.14 ±0.000 | 0.14 ±0.000 | 0.14 ±0.000 | 0.14 ±0.000 | 0.14 ±0.000 |
| | RealAWSCloudWatch | 0.18 ±0.000 | 0.18 ±0.000 | 0.18 ±0.000 | 0.18 ±0.000 | 0.18 ±0.000 | 0.18 ±0.000 |
| | ArtificialWithAnomaly | 0.18 ±0.000 | 0.18 ±0.000 | 0.18 ±0.000 | 0.18 ±0.000 | 0.18 ±0.000 | 0.18 ±0.000 |
| Case 3 | Synthetic Sines | 0.43 ±0.002 | 0.43 ±0.001 | 0.42 ±0.001 | 0.43 ±0.003 | 0.43 ±0.005 | 0.43 ±0.004 |
| | KPI | 0.33 ±0.003 | 0.33 ±0.004 | 0.32 ±0.001 | 0.32 ±0.009 | 0.32 ±0.006 | 0.32 ±0.005 |
| | RealTweets | 0.20 ±0.003 | 0.21 ±0.005 | 0.21 ±0.001 | 0.20 ±0.001 | 0.20 ±0.002 | 0.20 ±0.000 |
| | RealTraffic | 0.18 ±0.009 | 0.18 ±0.010 | 0.18 ±0.007 | 0.18 ±0.003 | 0.18 ±0.005 | 0.18 ±0.011 |
| | RealAWSCloudWatch | 0.19 ±0.001 | 0.19 ±0.002 | 0.19 ±0.002 | 0.19 ±0.000 | 0.19 ±0.001 | 0.19 ±0.003 |
| | ArtificialWithAnomaly | 0.19 ±0.004 | 0.19 ±0.001 | 0.19 ±0.008 | 0.19 ±0.002 | 0.19 ±0.003 | 0.19 ±0.001 |

Table 10: Comparison of change in detection performance under different training strategies.

| AD Algorithm | Dataset | w/o Aug. | C-GATS w/ end-to-end training | C-GATS w/ decoupled training |
|---|---|---|---|---|
| RobustTAD | Synthetic Sines | 0.74 ±0.005 | 0.74 ±0.002 | **0.75 ±0.004** |
| | KPI | 0.55 ±0.011 | 0.63 ±0.009 | **0.67 ±0.003** |
| | RealTweets | 0.54 ±0.021 | 0.56 ±0.011 | **0.57 ±0.010** |
| | RealTraffic | 0.67 ±0.026 | 0.70 ±0.021 | **0.73 ±0.015** |
| | RealAWSCloudWatch | 0.34 ±0.262 | 0.33 ±0.281 | **0.34 ±0.192** |
| | ArtificialWithAnomaly | 0.36 ±0.275 | 0.39 ±0.151 | **0.41 ±0.107** |
| SR-CNN | Synthetic Sines | 0.73 ±0.009 | 0.73 ±0.001 | **0.74 ±0.003** |
| | KPI | 0.54 ±0.008 | 0.57 ±0.004 | **0.58 ±0.007** |
| | RealTweets | 0.46 ±0.027 | 0.52 ±0.018 | **0.53 ±0.009** |
| | RealTraffic | 0.64 ±0.021 | 0.65 ±0.009 | **0.66 ±0.020** |
| | RealAWSCloudWatch | 0.25 ±0.319 | 0.25 ±0.220 | **0.27 ±0.199** |
| | ArtificialWithAnomaly | 0.28 ±0.019 | 0.33 ±0.015 | **0.34 ±0.011** |
| NCAD | Synthetic Sines | 0.77 ±0.008 | 0.77 ±0.007 | **0.79 ±0.001** |
| | KPI | 0.64 ±0.006 | 0.67 ±0.011 | **0.69 ±0.001** |
| | RealTweets | 0.57 ±0.019 | 0.60 ±0.021 | **0.61 ±0.009** |
| | RealTraffic | 0.69 ±0.029 | 0.71 ±0.012 | **0.71 ±0.019** |
| | RealAWSCloudWatch | 0.35 ±0.219 | 0.35 ±0.117 | **0.35 ±0.111** |
| | ArtificialWithAnomaly | 0.39 ±0.101 | 0.44 ±0.082 | **0.45 ±0.091** |

Table 11: Comparison of change in detection performance under different architecture choices.

| AD Algorithm | Dataset | w/o Aug. | C-GATS w/ conditioning at first layer | C-GATS w/ conditioning at last layer |
|---|---|---|---|---|
| RobustTAD | Synthetic Sines | 0.74 ±0.005 | 0.74 ±0.003 | **0.75 ±0.004** |
| | KPI | 0.55 ±0.011 | 0.60 ±0.007 | **0.67 ±0.003** |
| | RealTweets | 0.54 ±0.021 | 0.55 ±0.017 | **0.57 ±0.010** |
| | RealTraffic | 0.67 ±0.026 | 0.67 ±0.019 | **0.73 ±0.015** |
| | RealAWSCloudWatch | 0.34 ±0.262 | 0.32 ±0.198 | **0.34 ±0.192** |
| | ArtificialWithAnomaly | 0.36 ±0.275 | 0.36 ±0.119 | **0.41 ±0.107** |
| SR-CNN | Synthetic Sines | 0.73 ±0.009 | 0.74 ±0.002 | **0.74 ±0.003** |
| | KPI | 0.54 ±0.008 | 0.56 ±0.005 | **0.58 ±0.007** |
| | RealTweets | 0.46 ±0.027 | 0.48 ±0.011 | **0.53 ±0.009** |
| | RealTraffic | 0.64 ±0.021 | 0.65 ±0.019 | **0.66 ±0.020** |
| | RealAWSCloudWatch | 0.25 ±0.319 | 0.25 ±0.222 | **0.27 ±0.199** |
| | ArtificialWithAnomaly | 0.28 ±0.019 | 0.29 ±0.018 | **0.34 ±0.011** |
| NCAD | Synthetic Sines | 0.77 ±0.008 | 0.77 ±0.003 | **0.79 ±0.001** |
| | KPI | 0.64 ±0.006 | 0.66 ±0.010 | **0.69 ±0.001** |
| | RealTweets | 0.57 ±0.019 | 0.58 ±0.015 | **0.61 ±0.009** |
| | RealTraffic | 0.69 ±0.029 | 0.69 ±0.021 | **0.71 ±0.019** |
| | RealAWSCloudWatch | 0.35 ±0.219 | 0.35 ±0.109 | **0.35 ±0.111** |
| | ArtificialWithAnomaly | 0.39 ±0.101 | 0.41 ±0.159 | **0.45 ±0.091** |

