# OpenReview forum: "C-GATS: Conditional Generation of Anomalous Time Series"
_NeurIPS.cc/2022/Workshop/SyntheticData4ML — Neurips 2022 SyntheticData4ML_

### Official Review · Reviewer_3piW · 2022-10-16
**Empirical method for anomalous time series generation, many writing issues & questions about experiments**

**Rating:** 3
**Confidence:** 3

**Review:**

1. Summary
The authors describe a conditional generative model for anomalous time series.


2. Strengths and weaknesses.
Strengths:
- Important problem to tackle
- Good experimental setting, many trials including ablation studies, etc.

Weaknesses:
- the experimental results are a bit suspicious, as timeGAN and RCGAN perform quite badly.
From my experience, they should show decent performance in such tasks.
- some issues in the presentation



3. Questions
Could you explain the low performance of timeGAN and RCGAN in your experiments?

Please, attach the source code to supplementary materials. As the main contribution of this paper is that you provide a SOTA method, it is crucial to ensure reproducibility.


4. Limitations
I think the limitations are well discussed.


5. Issues.
- Figure 1 is too small; the text should be increased
- Figure 1: first image caption claims to show a vector, but it shows a function. Perhaps, it is better to use other visualization
- Figure 1: gaussian -> Gaussian
- L51-54: I don't think it is valid to use the curse of dimensionality analogy to the growth of y(t) length, as it is temporal
- L64 I guess the vast majority of synthetic data is downstream model agnostic. Maybe this claim is redundant
- L65 Also, I am a bit concerned about this one as a key contribution
- L78 VAEs do not suffer from mode collapse (authors support this claim by the reference to a GAN paper), and Figure 6 where there is no mode collapse. Perhaps, authors can investigate posterior collapse (which is a different thing!) in this case.
- Eq. 1 I guess instead $p(X^N$) it should be modeled as $p(X^N|Y^A)$, as the "distance from timepoint to the next anomaly can affect generation"
- L101-102: It is a theoretical part, but there are experimental details such as learning rate or specific architectures. I guess they can vary from dataset to dataset and should be included in the experimental section.
- use of the word "invariances": I think the word invariance is misused in this paper. Invariance is a property that stays the same under a set (or a group) of transformations. I don't think this method learns invariances (generate good synthetic data != learn invariances). Otherwise, it should be shown what invariances it learned.
- Tables 1 & 2: I think these tables could summarize better what the authors try to show (e.g., performance gain using synthetic data)
- Table 3: all best results should be highlighted, not only the author's model. For example, lines 1, 3, 7, ...
- Table 3: I am *very* surprised at how badly RCGAN works in these experiments. It contradicts my experience with it. It would be great to see the source code to verify that the RCGAN experiment is valid.
- MixUP, I don't think it is a good idea to apply mixup to this task (as it "averages out" the anomalies). Perhaps, it should be removed from the benchmarks.

7. Review
It is an important and interesting problem to tackle, and the method is quite nice (though it is not theoretically motivated). I like that the authors performed many experiments and investigated this method from various angles. The method is rather engineering, and IMO the theoretical part can be significantly improved (quite a lot of presentation and explanation issues, and even mistakes). The experimental results seem to be suspicious due to the very low performance of timeGAN and RCGAN. As the source code is not provided, and there are not enough reproducibility details, it cannot be checked.

---

### Official Review · Reviewer_3tBX · 2022-10-18
**C-GATS**

**Rating:** 7
**Confidence:** 3

**Review:**

The paper presents a method for time-series data generation based on variational autoencoders. The main goal of the method is to augment datasets for the downstream task of anomaly detection, which is usually hindered by unbalanced data. In order to insert anomalies into the GT data, they generate embedding representations of the time-series and perform a split of the embedding that allows them to independently operate on the normal and anomalous parts of the sequence.

CLARITY
* Section 3 is not very clear. There is a mixture between high-level explanation of the method, and low-level details. It would be better to include the diagram of the method in the paper, and refer to it across the method description.

PROS
* The paper is well written, although there are some frequent typos or grammatical errors that ideally should be fixed for camera-ready (revise for example the paragraph between lines 35 and 42).
* Includes a large number of additional materials that clarify the work.

CONS
* t-SNE evaluation is performed only on synthetic sines, and not real-world datasets.
* The "Mixup" method seems to be performing best in t-SNE evaluation. This should be discussed in the results.
* Not clear why results are better for real datasets than for synthetic one.
* Fourier Flow should be included as an additional generation method, given that is state-of-the-art currently.

---

### Official Review · Reviewer_GAsv · 2022-10-18
**Data augmentation for anomalous time series using a conditional generative model**

**Rating:** 6
**Confidence:** 3

**Review:**

Overall a simple but interesting method to augment time series used for anomaly detection.

Pros:
* Data generation for anomalous time series is a relevant problem with many useful applications.
* Good level of evaluation using several datasets, AD models and baselines.
* Paper is well written and easy to follow.

Comments:
* I'm not sure if Mixup and CutMix are good baselines for this problem.
* In Figure 2 I would like to see original anomalous time series to compare with generated ones.
* Figure 3 shows t-SNE visualizations of TSGAN, TimeGAN and cVAE which are not used in the quantitative results and are not discussed.
* I'm not sure if model agnostic and domain agnostic are contributions as for example all other methods used (especially combination of simple transformations) are model and to some extent domain agnostic.

---

### Official Review · Reviewer_w6rd · 2022-10-18
**Good paper with extensive experiments**

**Rating:** 8
**Confidence:** 3

**Review:**

In this paper, the authors present C-GATS, a semi-supervised approach to generate synthetic anomalous TS data with labels. This method is investigated empirically compared with 7 different TS generation/augmentation methods, and shows some promising results in both qualitative analysis and quantitative analysis.

Pros:

The paper is well written. The core idea is arrived at systematically and is carefully explained.

Compared to existing approaches, the proposed method is efficient and effective, achieving the state-of-the-art performance

The authors conduct extensive experiments to study and analyze the proposed method

Cons:

I did not see any major problems with the paper.

---

### Meta-Review · Area_Chair_RXW9 · 2022-10-19

**Recommendation:** Accept

**Review:**

The authors propose a method to generate anomalous TS with labels. The reviewers all agree on the importance of the problem and the quality of the proposed method, and 3 of the reviewers emphasize the quality of the experimental section, thus I suggest an accept for this paper. However, the authors should take into account the suggestions regarding the baselines (adding more of them), and providing an explanation for why methods such as RCGAN do not work well, forthe camera-ready version.